# Learning from Future: A Novel Self-Training Framework for Semantic Segmentation

**Ye Du**[1,2]    **Yujun Shen**[3]    **Haochen Wang**[4]    **Jingjing Fei**[5]    **Wei Li**[5]
**Liwei Wu**[5]    **Rui Zhao**[5,6]    **Zehua Fu**[1,2]    **Qingjie Liu**[1,2*]

[1] State Key Laboratory of Virtual Reality Technology and Systems, Beihang University
[2] Hangzhou Innovation Institute, Beihang University
[3] The Chinese University of Hong Kong
[4] Institute of Automation, Chinese Academy of Sciences    [5] SenseTime Research[†]
[6] Qing Yuan Research Institute, Shanghai Jiao Tong University, Shanghai, China

{duyee, zehua_fu, qingjie.liu}@buaa.edu.cn    shenyujun0302@gmail.com
wanghaochen2022@ia.ac.cn    {feijingjing1, liwei1, wuliwei, zhaorui}@sensetime.com

## Abstract

Self-training has shown great potential in semi-supervised learning. Its core idea is to use the model learned on labeled data to generate pseudo-labels for unlabeled samples, and in turn teach itself. To obtain valid supervision, active attempts typically employ a momentum teacher for pseudo-label prediction yet observe the confirmation bias issue, where the incorrect predictions may provide wrong supervision signals and get accumulated in the training process. The primary cause of such a drawback is that the prevailing self-training framework acts as guiding the current state with previous knowledge, because the teacher is updated with the past student only. To alleviate this problem, we propose a novel self-training strategy, which allows the model to *learn from the future*. Concretely, at each training step, we first virtually optimize the student (*i.e.*, caching the gradients without applying them to the model weights), then update the teacher with the virtual future student, and finally ask the teacher to produce pseudo-labels for the current student as the guidance. In this way, we manage to improve the quality of pseudo-labels and thus boost the performance. We also develop two variants of our *future-self-training* (FST) framework through peeping at the future both deeply (FST-D) and widely (FST-W). Taking the tasks of unsupervised domain adaptive semantic segmentation and semi-supervised semantic segmentation as the instances, we experimentally demonstrate the effectiveness and superiority of our approach under a wide range of settings. Code is available at https://github.com/usr922/FST.

## 1 Introduction

Improving the labeling efficiency of deep learning algorithms is vital in practice since acquiring high-quality annotations could consume great effort. Self-training (ST) offers a promising solution to alleviate this issue by learning with limited labeled data and large-scale unlabeled data [47, 24]. The key thought is to learn a model on labeled samples and use it to generate pseudo-labels for unlabeled samples to teach the model itself. In general, a teacher network that maintains an exponential moving average (EMA) of the student (*i.e.*, the model to learn) weights is used for pseudo-label prediction, as shown in Fig. 1a. Intuitively, such a training strategy relies on the *previous* student states to supervise

---

[*]Corresponding Author.

[†]This work is done when Ye Du and Haochen Wang are interns at SenseTime Research.

36th Conference on Neural Information Processing Systems (NeurIPS 2022).

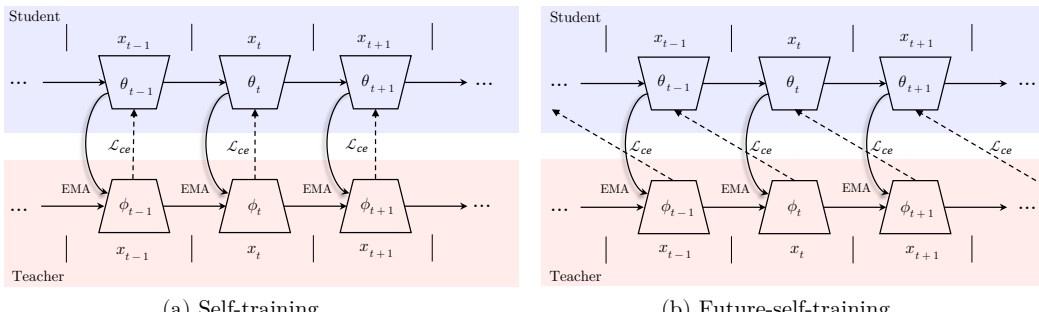

(a) Self-training          (b) Future-self-training

Figure 1: **Concept comparison** between self-training (ST) and our future-self-training (FST). (a) ST employs a teacher, which collects information from the *past* states, to supervise the student. (b) Our FST derives a teacher at the *future* moment and utilizes it to guide the current student.

the current state, which amounts to using a poor model to guide a good one given the fact that a model tends to perform better along with the training process. As a result, the confirmation bias issue [4, 9] emerges from existing ST approaches, where the wrong supervision signals caused by those incorrect pseudo-labels get accumulated during training.

To break through the predicament of seeking supervision only from the past states, we propose *future-self-training* (FST), which allows the model to learn from its *future self*. Fig. 1b illustrates the concept diagram of our FST. Compared to the conventional ST framework in Fig. 1a, which employs the $t$-step teacher (*i.e.*, updated with the student at moments $1, 2, \ldots, t-1$) to guide the $t$-step student, FST presents a new training manner by urging the $t$-step student to learn from the $(t+1)$-step teacher. However, at the start of the training step $t$, the $(t+1)$-step teacher is not available yet since it is dependent on the to-be-optimized $t$-step student. To tackle this obstacle, we come up with a *virtual updating* strategy. Concretely, we first optimize the current student just like that in the traditional ST. Differently, we do *not* actually update the student weights but cache the gradients instead. Such stashed gradients can be treated as the "virtual future" and help derive the $(t+1)$-step teacher. Finally, the training of step $t$ borrows the pseudo-labels predicted by the latest teacher, and this time we apply the gradients to the student weights for real.

Recall that our motivation of encouraging the model to learn from the future is to help it acquire knowledge from an advanced teacher. To this end, we put forward two variants based on our FST framework to make the teacher more capable. On the one hand, we propose FST-D to investigate the future *deeply*. For this case, we ask the teacher to move forward for $K$ steps via virtual updating, thus the $t$-step student can be better supervised by the $(t+K)$-step teacher. On the other hand, FST-W originates from the idea of model soups [53], which reveals that the averaging weights of multiple fine-tuned models can improve the performance. We hence propose to explore the future *widely* with teachers developed from different training samples and expect the student to learn from all these $(t+1)$-step teachers simultaneously.

We evaluate our proposed FST on the tasks of both unsupervised domain adaptive (UDA) semantic segmentation and semi-supervised semantic segmentation. The superiority of FST over the prevailing ST framework is summarized in Fig. 2, where our teacher model is capable of producing pseudo-labels with much higher quality and hence assists the student with a better performance. This is because, along with the training process, the future states usually outperform the past states and thus can provide more accurate supervision, reducing the damage of confirmation bias. Such a comparison validates our primary motive of learning from the future. Furthermore, we

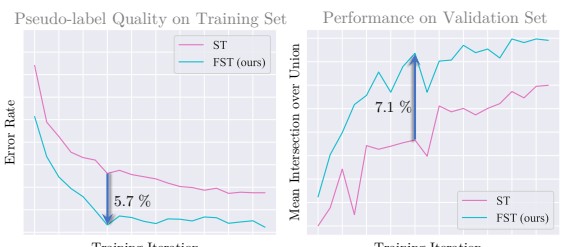

Figure 2: **Performance comparison** between self-training (ST) and our future-self-training (FST), including the pseudo-label quality on unlabeled training samples (left) and the evaluation performance (right). The comparison is conducted under the same number of updates of the student, which is the final model used for evaluation.

observe consistent performance gain under a broad range of experimental settings (*e.g.*, network architectures and datasets), demonstrating the effectiveness and generalizability of our approach.

## 2 Related work

**Domain adaptive semantic segmentation.** UDA semantic segmentation aims at transferring the knowledge from a labeled source domain to an unlabeled target domain, which is often viewed as a special semi-supervised learning problem. Early methods for UDA segmentation focus on diminishing the distribution shift between the source and target domain at the input level [23, 40, 18], the feature level [46, 10, 8, 31], or the output level [46, 48, 36]. Over the years, adversarial learning [19, 17] has been the dominant approach to aligning the distributions. However, the alignment-based methods may destroy the discrimination ability of features and cannot guarantee a small expected error on the target domain [58]. In contrast, self-training [2], which is originated from semi-supervised learning (SSL) [29], is introduced to directly minimize a proxy cross-entropy (CE) loss on the target domain. By leveraging the model itself to generate pseudo-labels on unlabeled data, self-training together with tailored strategies such as consistency regularization [61, 3], cross-domain mixup [45, 62], contrastive learning [26, 34, 63, 31], pseudo-label refine [51, 58, 60], auxiliary tasks [49, 50] and class balanced training [30] achieves excellent performance. Recently, Hoyer et al. [24] empirically proved that the transformer architecture [55] is more robust to domain shift than CNN. They propose a transformer-based framework with three efficient training strategies in pursuit of milestone performance.

**Semi-supervised semantic segmentation.** Self-training is widely studied in SSL literature [47]. To facilitate the usage of unlabeled samples, Tarvainen et al. [44] propose a mean teacher framework for consistency learning between a *student* and a momentum updating *teacher*. This idea is extended later to semi-supervised semantic segmentation, which trains the student model with high-confident *hard* pseudo-labels predicted by the teacher. On this basis, extensive attempts improve semi-supervised semantic segmentation by CutMix augmentation [16], class-balanced training [64, 25, 21] and contrastive learning [64, 1, 33, 52]. A closely relevant topic to self-training in SSL is consistency regularization, which believes that enforcing semantic or distribution consistency between various perturbations, such as image augmentation [27] and network perturbation [57], can improve the robustness and generalization of the model. In general, consistent regularization methods are used together with a ST framework. We focus on improving the basic ST in this work.

**Nesterov's accelerated gradient descent.** A related idea to our work is Nesterov's accelerated gradient descent (NAG). Originally proposed in [37] for solving convex programming problem, NAG is a first-order optimization method with a better convergence rate than gradient descent. With the rise of deep learning, NAG is adopted as an alternative to momentum stochastic gradient descent (SGD) to optimize neural networks [43, 14]. It is intuitively considered to perform a look ahead gradient evaluation and then make a correction [7]. Due to its solid theoretical explanations [65, 5] and remarkable performance, many works incorporate NAG with various tasks. In [32], Lin et al. adopt NAG into the area of adversarial attack, where they propose a Nesterov's iterative fast gradient sign method to improve the transfer ability of adversarial examples. In [56], Yang et al. explore the utilization of NAG in federal learning. Different from NAG that pursues accelerated convergence, our work aims at building a stronger pseudo-label generator and improving the performance of traditional self-training.

## 3 Method

### 3.1 Background

Consider such a real-world scenario where we have access to a labeled segmentation dataset $\mathcal{D}_L = \{x_l, y_l\}_{l=1}^{n_l}$ from distribution $P$ and an unlabelled one $\mathcal{D}_U = \{x_u\}_{u=1}^{n_u}$ from unknown distribution $Q$. We are required to build a semantic segmentation model using the combination of $\mathcal{D}_L$ and $\mathcal{D}_U$. A general case is when $P \neq Q$, the problem falls into the category of UDA semantic segmentation. Otherwise, it is usually treated as a regular SSL task.

Self-training provides a unified solution and achieves state-of-the-art performance on both settings [24, 52]. One of the most common and widely used forms of self-training in semantic segmentation is a variant of mean teacher, which is shown in Fig. 3. Denote by $g_\theta$ the segmentation model required to be trained, and $\theta$ its parameters. The mean teacher framework trains the *student* $g_\theta$ on unlabeled data with pseudo-labels predicted by a momentum *teacher* $g_\phi$, which has the same architecture to the student but with different parameters $\phi$. Specifically, as the training progresses, the teacher

evolves with the student by maintaining an EMA of student weights on each training iteration. This ensembling enables generating high quality predictions on unlabeled samples, and using them as training targets improves performance. Formally, at each training step, the teacher is first updated and then predict pseudo-labels to train the student.

$$\begin{aligned} \phi_{t+1} &= \mu\phi_t + (1-\mu)\theta_t, \\ \theta_{t+1} &= \theta_t - \gamma\nabla_\theta\left[\mathcal{L}(g_{\theta_t}(x_l), y_l) + \lambda\mathcal{L}(g_{\theta_t}(x_u), \hat{y}_u|\phi_{t+1})\right], \end{aligned} \quad (1)$$

where $\mu$ is the momentum coefficient, $\gamma$ is the learning rate, and $\lambda$ is the dynamic re-weighting parameter to weigh the training of labeled and unlabeled data. $\hat{y}_u$ denotes the pseudo-labels predicted by $\phi_{t+1}$, *i.e.*, $\hat{y}_u = \arg\max g_{\phi_{t+1}}(x_u)$. $\mathcal{L}$ is the pixel-wise cross-entropy training objective, which can be written as

$$\mathcal{L}(x,y) = -\sum_{j=1}^{H\times W}\sum_{c=1}^{C}\mathbb{I}_{y^{j,c}=1}\log g_\theta(x)^{j,c}, \quad (2)$$

where $H \times W$ is the input image size and $C$ is the total number of classes.

**Limitation of self-training.** Despite the remarkable performance, self-training suffers from the problem of confirmation bias. To be specific, the inherent noise in pseudo-labels could undesirably mislead the student training, which in return affects the pseudo-label prediction, and thereby results in noise accumulation. Though a momentum updating strategy in the mean teacher framework improves tolerance with inaccurate pseudo-labels, this issue is still a bottleneck since the student still relies on learning from its own *past* training states.

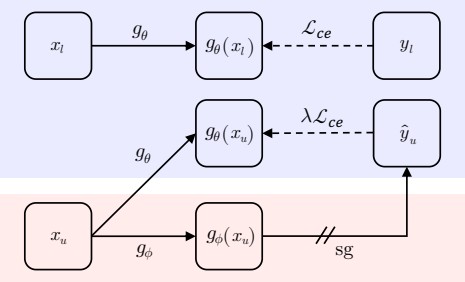

Figure 3: **Illustration** of the ST framework with a teacher $g_\phi$. "sg" means stop-gradient.

## 3.2 Learning from future self

An intuitive observation shown in Fig. 2 is that the performance of the student model generally improves during training, despite the noise in supervision. From this perspective, a reasonable conjecture is, can we use model information from future moments to guide the current training iteration? Motivated by this, we propose *future-self-training* for facilitating the utilization of unlabeled data in semantic segmentation. Concretely, at each training step, we propose to directly update the teacher model by the student weights from the next training moment. To this end, a simple modification to Eq. (1) is made as follows.

$$\begin{aligned} \phi_{t+1} &= \mu\phi_t + (1-\mu)\left(\theta_t - \gamma\nabla_\theta\left[\mathcal{L}(g_{\theta_t}(x_l), y_l) + \lambda\mathcal{L}(g_{\theta_t}(x_u), \hat{y}_u|\phi_t)\right]\right), \\ \theta_{t+1} &= \theta_t - \gamma\nabla_\theta\left[\mathcal{L}(g_{\theta_t}(x_l), y_l) + \lambda\mathcal{L}(g_{\theta_t}(x_u), \hat{y}_u|\phi_{t+1})\right]. \end{aligned} \quad (3)$$

Furthermore, it can be seen that Eq. (3) only uses a virtual future state to update the teacher and ignores the current student weights $\theta_t$. Our mission here is to establish a reliable pseudo-label generator (*i.e.* a stronger teacher). In terms of the ensembling effect of EMA, it is not necessary to discard $\theta_t$. Therefore, an improved version of FST is proposed as follows.

$$\begin{aligned} \phi'_{t+1} &= \mu\phi_t + (1-\mu)\theta_t, \\ \phi_{t+1} &= \mu'\phi'_{t+1} + (1-\mu')(\theta_t - \gamma\nabla_\theta[\mathcal{L}(g_{\theta_t}(x_l), y_l) + \lambda\mathcal{L}(g_{\theta_t}(x_u), \hat{y}_u|\phi'_{t+1})]), \\ \theta_{t+1} &= \theta_t - \gamma\nabla_\theta\left[\mathcal{L}(g_{\theta_t}(x_l), y_l) + \lambda\mathcal{L}(g_{\theta_t}(x_u), \hat{y}_u|\phi_{t+1})\right], \end{aligned} \quad (4)$$

where a new momentum parameter $\mu'$ is introduced to distinguish the contribution of current and future model weights to teacher updates. We provide pseudo-codes to further illustrate how we implement Eq. (4) in *Supplementary Material*.

## 3.3 Exploring a deeper future

We reiterate that the key insight of FST is to look ahead during training, which allows to mine more accurate supervision from future model states. In experiments, we found that Eq. (4) exhibits only a slight improvement in performance (Tab. 2), showing that this one-step future exploration strategy is insufficient.

Therefore, we further propose a looking ahead *deeper* strategy to peek into deeper future student states. To be specific, at each training step, we update the teacher not only with the student weights from the next moment, but also with those from deeper steps. Formally, denote by $\widetilde{\phi}_t = \mu\phi_t + (1-\mu)\theta_t$ and $\widetilde{\theta}_t = \theta_t$ two agent variables firstly. Then, we can use the co-evolving $\widetilde{\phi}_t$ and $\widetilde{\theta}_t$ for *virtual updating* as follows.

$$\widetilde{\theta}_{t+k+1} = \widetilde{\theta}_{t+k} - \gamma\nabla_{\widetilde{\theta}}[\mathcal{L}(g_{\widetilde{\theta}_{t+k}}(x_l), y_l) + \lambda\mathcal{L}(g_{\widetilde{\theta}_{t+k}}(x_u), \hat{y}_u|\widetilde{\phi}_{t+k})],$$
$$\widetilde{\phi}_{t+k+1} = \mu'\widetilde{\phi}_{t+k} + (1-\mu')(\widetilde{\theta}_{t+k+1}), \tag{5}$$

where $k = \{0, ..., K-1\}$ indexes the serial virtual steps for current training and $K$ is the total number of exploration steps. Finally, we use the future information aware teacher $\widetilde{\phi}_{t+K}$ as the pseudo-label generator to supervise the current training. A simple reassignment and gradient descent update are applied to form the deeper version of FST, which is so called FST-D as shown below.

$$\phi_{t+1} = \widetilde{\phi}_{t+K},$$
$$\theta_{t+1} = \theta_t - \gamma\nabla_{\theta}[\mathcal{L}(g_{\theta_t}(x_l), y_l) + \lambda\mathcal{L}(g_{\theta_t}(x_u), \hat{y}_u|\phi_{t+1})]. \tag{6}$$

## 3.4 Exploring a wider future

On the other hand, looking ahead *wider* instead of *deeper* is another intuitive way to enhance future exploration. Inspired by the recent progress [53] that an ensemble of different model weights often shows excellent performance, we propose to first explore the next moment in different optimization directions and then use the average of them to update the teacher. Concretely, we obtain different optimization directions by feeding *different data batches* to the student model at each training moment. Thus, a wider version of FST, *i.e.*, FST-W, is presented as follows.

$$\phi_{t+1} = \mu'\{\mu\phi_t + (1-\mu)\theta_t\} + (1-\mu')(\theta_t - \frac{1}{N}\sum_{i=1}^{N}\gamma\nabla_{\theta}[\mathcal{L}(g_{\theta_t}(x_l^i), y_l^i) + \lambda\mathcal{L}(g_{\theta_t}(x_u^i), \hat{y}_u^i|\phi_t)]),$$
$$\theta_{t+1} = \theta_t - \gamma\nabla_{\theta}[\mathcal{L}(g_{\theta_t}(x_l), y_l) + \lambda\mathcal{L}(g_{\theta_t}(x_u), \hat{y}_u|\phi_{t+1})], \tag{7}$$

where $i$ indexes different samples and $N$ is parallel virtual exploration steps.

Eq. (7) holds due to the fact that averaging the model weights is equivalent to averaging the gradients first and then updating the parameters by gradient descent. It is worth noting that FST-D and FST-W are complementary that can be utilized together. However, this is beyond the scope of our work, and we leave this exploration to the future.

## 4 Experiment

The experiment section is organized as follows. First, we illustrate the experimental setup and implementation details in Sec. 4.1 and Sec. 4.2. Then, we evaluate the proposed FST and analyze the two variants in Sec. 4.3. After that, we conduct extensive ablation studies to dissect our method in Sec. 4.4. Finally, we compare our FST with existing state-of-the-art alternatives on both UDA and semi-supervised benchmarks in Sec. 4.5.

## 4.1 Setup

**Datasets and tasks.** We evaluated our method on UDA and semi-supervised semantic segmentation. In UDA segmentation, we use synthetic labeled images from GTAV [38] and SYNTHIA [39] as the source domain and use real images from Cityscapes [13] as the target domain. In addition, PASCAL VOC 2012 [15] is used for standard semi-supervised evaluation. To simulate a semi-supervised setting, we randomly sample a portion *(i.e.*, 1/4, 1/8, and 1/16) of images together with corresponding segmentation masks from the training set as the labeled data and treat the rest as the unlabeled samples.

**Evaluation metric.** Mean Intersection over Union (mIoU) is reported for evaluation. In SYNTHIA $\rightarrow$ Cityscapes UDA benchmark, 16 and 13 of the 19 classes of Cityscapes are used to calculate mIoU, following the common practice [3, 24].

Table 1: **Comparison between ST and our FST**, where we explore the future with either (a) the same data batch as the current or (b) a different data batch from the current. "SourceOnly" means training the model with labeled data only, whose result is borrowed from [24] as the reference. $4\times$ means using quadruple samples per mini-batch. All results are averaged over 3 random seeds.

| Method | mIoU | $\Delta$ |
|---|---|---|
| SourceOnly | $34.3 \pm 2.2$ | - |
| ST | $56.3 \pm 0.4$ | - |
| - | - | - |
| Naive-FST | $56.4 \pm 0.4$ | $\uparrow 0.1$ |
| Improved-FST | $57.7 \pm 0.6$ | $\uparrow 1.4$ |
| FST-W | $56.8 \pm 0.1$ | $\uparrow 0.5$ |
| FST-D | $\mathbf{59.8 \pm 0.1}$ | $\uparrow \mathbf{3.5}$ |

(a) Future exploration with the same data batch.

| Method | Batch | mIoU | $\Delta$ |
|---|---|---|---|
| SourceOnly | $1\times$ | $34.3 \pm 2.2$ | - |
| ST | $1\times$ | $56.3 \pm 0.4$ | - |
| ST | $4\times$ | $55.5 \pm 0.4$ | $\downarrow 0.8$ |
| Naive-FST | $1\times$ | $58.7 \pm 2.3$ | $\uparrow 2.3$ |
| Improved-FST | $1\times$ | $58.7 \pm 0.7$ | $\uparrow 2.4$ |
| FST-W | $1\times$ | $59.3 \pm 0.5$ | $\uparrow 3.0$ |
| FST-D | $1\times$ | $\mathbf{59.6 \pm 1.4}$ | $\uparrow \mathbf{3.3}$ |

(b) Future exploration with a different data batch.

**Baselines.** We first build strong baselines of the classical ST framework. For UDA segmentation, we adopt the basic framework from [45], which contains a ClassMix augmentation. Standard cross-entropy loss is calculated on both labeled and unlabeled data. We use the efficient Encoder-Decoder structure for all semantic segmentation models, where the networks various in the structure of encoders and decoders. In the semi-supervised benchmark, we use the classical ST without other tricks as the baseline, because it has been proved to achieve competitive performance while maintaining simplicity [27].

### 4.2 Implementation details

**Image augmentation**. The proposed FST and its baselines use the same image augmentation for fair comparison. In UDA semantic segmentation, color jitter, Gaussian blur and ClassMix [45] are used as the strong data augmentation for the unlabeled target domain, which follows the practice in [24]. In semi-supervised semantic segmentation, we use random flip and random crop, and the images are resized to $513 \times 513$ for both teacher and student.

**Network architecture**. We use the DeepLabV2 [11] as the basic segmentation architecture for UDA segmentation, where the ASPP decoder only uses the dilation rates 6 and 12 following [46]. For Transformer-based networks, we adopt from [24] and [54] as the decoders. In semi-supervised segmentation, we evaluate our method on the commonly used DeepLabV2 [22], DeepLabV3+ [12] and PSPNet [59] with ResNet-101 [22] as the backbone.

**Optimization**. In UDA segmentation, the model is trained with an AdamW [28] optimizer, a learning rate of $6 \times 10^{-5}$ for the encoder and $6 \times 10^{-4}$ for the decoder, a weight decay of $0.01$, linear learning rate warmup with 1.5k iterations and linear decay afterwards. We train the model on a batch of two $512 \times 512$ random crops for a total of 40k iterations. The momentum $u$ is set to $0.999$. In semi-supervised segmentation, the model is trained with a SGD optimizer, a learning rate of $0.0001$ for the encoder and $0.001$ for the decoder, a weight decay of $0.0001$. We train the model with 16 labeled and 16 unlabeled images per-batch for a total of 40 epochs.

### 4.3 Comparison with self-training

We first comprehensively compare our FST with classical ST to evaluate the effectiveness. The results are shown in Tab. 1. To simplify, we use GTAV as the labeled data and Cityscapes as the unlabeled data for evaluation. All methods use the same experimental settings for fairness.

**Quantitative analyses.** We illustrate the improvements of Naive-FST (Eq. (3)), Improved-FST (Eq. (4)), FST-D (Eqs. (5) and (6)) and FST-W (Eq. (7)) compared with classical ST (Eq. (1)) in Tab. 1a. These methods use the same batch of data for virtual forward at each step of future exploration. As presented, Naive-FST only shows a negligible boost because the current student state is discarded without contributing to the teacher. By revising it, the improved FST in Eq. (6), which is a special case of FST-D when $K = 1$, achieves an improvement of $1.4\%$ mIoU. Further, FST-D (with $K = 3$) clearly outperforms ST by a margin of $3.5\%$ mIoU, which benefits from the higher-quality pseudo-labels generated by a more reliable teacher as shown in Fig. 2. In contrast, FST-W shows a slight improvement of only $0.5\%$ mIoU under the same data batch setting. Thus, we

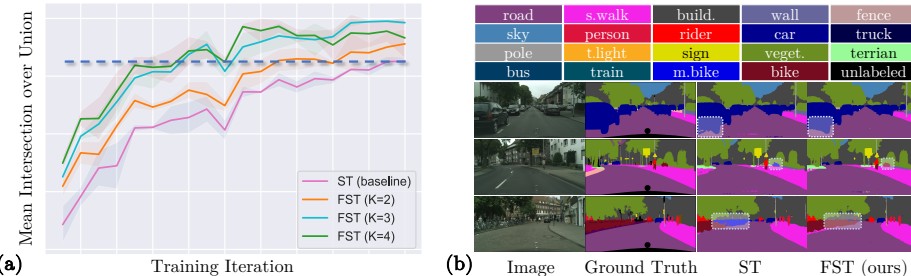

Figure 4: (a) **Performance curves** for ST and FST with various $K$ values. The comparison is conducted under the same number of updates of the student, which is the final model used for evaluation. (b) **Qualitative comparison** on Cityscapes [13], where dashed white boxes highlight the visual improvements.

Table 2: **Generalization** of FST across architectures. All results are averaged over 3 random seeds.

| Method | $K$ | mIoU | $\Delta$ | Method | $K$ | mIoU | $\Delta$ | Method | $K$ | mIoU | $\Delta$ |
|---|---|---|---|---|---|---|---|---|---|---|---|
| ST | - | $55.0 \pm 0.9$ | - | ST | - | $56.3 \pm 0.4$ | - | ST | - | $56.3 \pm 0.8$ | - |
| FST | 2 | $56.3 \pm 1.0$ | ↑ 1.3 | FST | 2 | $57.8 \pm 1.3$ | ↑ 1.5 | FST | 2 | $58.1 \pm 3.1$ | ↑ 1.8 |
| FST | 3 | $\mathbf{56.9 \pm 0.5}$ | ↑ **1.9** | FST | 3 | $\mathbf{59.8 \pm 0.1}$ | ↑ **3.5** | FST | 3 | $58.5 \pm 0.7$ | ↑ 2.2 |
| FST | 4 | $56.4 \pm 0.9$ | ↑ 1.4 | FST | 4 | $59.7 \pm 0.8$ | ↑ 3.4 | FST | 4 | $\mathbf{58.8 \pm 1.0}$ | ↑ **2.5** |

(a) DeepLabV2 [11] *w/* ResNet-50 [22].    (b) DeepLabV2 [11] *w/* ResNet-101 [22].    (c) PSPNet [59] *w/* ResNet-101 [22].

| Method | $K$ | mIoU | $\Delta$ | Method | $K$ | mIoU | $\Delta$ | Method | $K$ | mIoU | $\Delta$ |
|---|---|---|---|---|---|---|---|---|---|---|---|
| ST | - | $61.3 \pm 0.7$ | - | ST | - | $59.9 \pm 2.0$ | - | ST | - | $68.3 \pm 0.5$ | - |
| FST | 2 | $63.7 \pm 2.0$ | ↑ 2.4 | FST | 2 | $62.5 \pm 1.2$ | ↑ 2.6 | FST | 2 | $69.1 \pm 0.3$ | ↑ 0.8 |
| FST | 3 | $64.3 \pm 2.3$ | ↑ 3.0 | FST | 3 | $62.5 \pm 1.9$ | ↑ 2.6 | FST | 3 | $\mathbf{69.3 \pm 0.3}$ | ↑ **1.0** |
| FST | 4 | $\mathbf{64.4 \pm 2.0}$ | ↑ **3.1** | FST | 4 | $\mathbf{62.6 \pm 1.8}$ | ↑ **2.7** | FST | 4 | $68.8 \pm 0.9$ | ↑ 0.5 |

(d) UPerNet [54] *w/* Swin-B [35].    (e) UPerNet [54] *w/* BEiT-B [6].    (f) DAFormer [24] *w/* MiT-B5 [55].

prefer the deeper variant and adopt it as the basic approach in this paper, *i.e.*, FST stands for FST-D unless specified. We also analyse the effect of exploration steps (*i.e.*, $K$) on the training process. As suggested in Fig. 4a, FST spends only about $1/3$ of the total training time to reach the performance level of ST. Besides, we find that a larger $K$ can achieve higher mIoU at the beginning of the training process. When $K = 4$, however, the performance in later training iterations drops and gets worse than $K = 3$. We speculate that this is because the deeper exploration becomes unnecessary in the later training stage. This interesting phenomenon indicates that an adaptive exploration mechanism may bring better results.

**Qualitative analyses.** Fig. 4b provides some qualitative comparisons, where our FST can correct some mistakes made by ST. Taking the presented sample in the second row as an instance, ST struggles to distinguish between *bicycle* and *motorcycle*, while our FST successfully predicts it. More visualization results and analyses can be found in *Supplementary Material*.

**Data batches for future exploration.** In Sec. 3.4, we derive FST-W, which uses different samples for future exploration in parallel. Tab. 1a and Tab. 1b compare the performance of using the same and different data batches. Note that FST-W in Tab. 1a could produce slightly different mixed images for virtual forward, since we use ClassMix augmentation. It is obvious that the parallel exploration with different samples performs better because the differences between models are important for ensembling.

**Generalization of popular architectures**. To verify the generality under various advanced semantic segmentation models, we evaluate FST (the deeper variant) on two mainstream backbones (*i.e.*, CNN and Transformer) with four commonly used segmentation decoders. As presented in Tab. 2, FST shows consistent performance improvement over classical ST, including DeepLab [11], PSPNet [59] and UPerNet [54]. Besides, FST shows significant improvements not only on supervised pretrained CNN [22] and Transformer backbones [35, 55] but also on unsupervised pretrained BEiT [6]. Note that, the established ST baselines are strong, which even surpass many complex multi-stage methods (*e.g.*, [58]) proposed recently. FST achieves $59.8\%$ mIoU using DeepLabV2 and ResNet-101. More comparisons between FST and existing CNN-based methods are provided in *Supplementary Material*.

Table 3: **Analyses on the two variants of FST**, including FST-D (Sec. 3.3) and FST-W (Sec. 3.4). All results are averaged over 3 random seeds.

| Method | Backbone | $K$ | mIoU | $\Delta$ |
|---|---|---|---|---|
| ST | ResNet-101 | - | $56.3 \pm 0.4$ | - |
| FST-D | ResNet-101 | 2 | $58.6 \pm 0.4$ | $\uparrow 2.3$ |
| FST-D | ResNet-101 | 3 | $59.6 \pm 1.4$ | $\uparrow 3.3$ |
| FST-D | ResNet-101 | 4 | $\mathbf{59.8 \pm 2.0}$ | $\uparrow \mathbf{3.5}$ |

(a) Effect of $K$ in FST-D.

| Method | Backbone | $N$ | mIoU | $\Delta$ |
|---|---|---|---|---|
| ST | ResNet-101 | - | $56.3 \pm 0.4$ | - |
| FST-W | ResNet-101 | 2 | $58.5 \pm 1.6$ | $\uparrow 2.2$ |
| FST-W | ResNet-101 | 3 | $\mathbf{59.3 \pm 0.5}$ | $\uparrow \mathbf{3.0}$ |
| FST-W | ResNet-101 | 4 | $58.6 \pm 2.0$ | $\uparrow 2.3$ |

(b) Effect of $N$ in FST-W.

Table 4: (a) **Ablation study** on the hyper-parameter $\mu'$ (Sec. 3.2). (b) **Comparison with longer-training baselines.** All results are averaged over 3 random seeds.

| Method | $\mu'$ | mIoU | $\Delta$ |
|---|---|---|---|
| ST | - | $56.3 \pm 0.4$ | - |
| - | - | - | - |
| FST | 0.99 | $58.8 \pm 1.6$ | $\uparrow 2.5$ |
| FST | 0.999 | $59.8 \pm 0.1$ | $\uparrow 3.5$ |
| FST | 0.9999 | $58.7 \pm 0.6$ | $\uparrow 2.4$ |
| FST | 0.99999 | $\mathbf{59.9 \pm 0.9}$ | $\uparrow \mathbf{3.6}$ |

(a) Effect of $\mu'$ in Eq. (5).

| Method | Backbone | Schedule | mIoU | $\Delta$ |
|---|---|---|---|---|
| ST | ResNet-101 | $1\times$ | $56.3 \pm 0.4$ | - |
| ST | ResNet-101 | $4\times$ | $59.3 \pm 0.6$ | $\uparrow 3.0$ |
| FST | ResNet-101 | $1\times$ | $\mathbf{59.8 \pm 0.1}$ | $\uparrow \mathbf{3.5}$ |
| ST | MiT-B5 | $1\times$ | $68.3 \pm 0.5$ | - |
| ST | MiT-B5 | $3\times$ | $68.3 \pm 1.1$ | $\uparrow 0.0$ |
| FST | MiT-B5 | $1\times$ | $\mathbf{69.1 \pm 0.3}$ | $\uparrow \mathbf{0.8}$ |

(b) Comparison with longer training schedules.

## 4.4 Ablation studies

**Deeper or wider.** It can be seen from Tab. 1 that FST-D performs better than FST-W no matter using the same or different data batches for future exploration. It is worth noting that under the setting of Tab. 1b, the teacher model in FST sees more data per-iteration compared to the original ST, which has the effect of expanding the batch size in disguise. To make a fair comparison, we also build a ST baseline with a larger batch size. The results show that the performance gain of FST-W comes from the method itself instead of utilizing more data in each iteration. In addition, we conduct ablations to compare FST-D and FST-W with each step (*i.e.*, $K$ and $N$), which are shown in Tab. 3a and Tab. 3b. Both variants use different data batches for future exploration since FST-W performs well only under this setting. It can be concluded that FST-D performs better than FST-W, which is consistent with the conclusion in the above. Besides, comparing Tab. 2b and Tab. 3a, we observe that using different data batches amplifies the performance jitter of each run. We guess that this may be due to the diversity of data sampled for future exploration.

**Effect of the serial exploration steps $K$.** $K$ controls the number of virtual exploration steps in FST-D. In Tab. 2, we ablate $K$ on six settings, each evaluated over 3 runs. As presented, $K = 3$ shows steady improvements, while increasing it further brings negligible impact. Thus, we recommend using $K = 3$ as the basic practice. We also ablate $K$ when using different data batches for exploration, which is presented in Tab. 3a.

**Effect of the momentum $\mu'$.** It is a common practice to set the momentum $\mu$ of EMA to a large value such as 0.999 in self-training. A separate momentum $\mu'$ that controls the contribution of future student states to the teacher is set in our FST. We conduct ablation experiments to observe the effect of $\mu'$. As shown in Tab. 4a, FST shows robustness against the change of $\mu'$. In our experiments, we set $\mu' = 0.999$ as the default setting unless specifically stated, which equals the value of $\mu$.

**Effect of the parallel exploration steps $N$.** In FST-W, $N$ controls the parallel exploration steps of the next training moment. We conduct ablation experiments in Tab. 3b to verify the influence of $N$. As can be seen, $N = 3$ performs well among the evaluated values, which is a similar observation to $K$ and implies that it is an acceptable choice in practice.

**Longer training schedules.** We perform forward and backward propagation to obtain weights as the estimation of future student states. This simple implementation linearly increases the training time over the number of exploration steps, *i.e.*, $K$. Note that in our method, the student is trained with the same training iterations as classical ST and does *not* see more samples per-iteration, thereby the comparisons in Tabs. 1 and 2 are totally fair. Even though, we establish stronger baselines with longer training schedules and compare them with our method. We find that the performance of the longer scheduled ST baseline decreases heavily in the later training stages as the model is fitting the

Table 5: **Evaluation on the semi-supervised learning (SSL) setting** on PASCAL VOC 2012 [15], where 1/16, 1/8, and 1/4 stand for using 664, 1323, and 2646 samples as the labeled set, respectively.

| Method | PSPNet [59] | | | DeepLabV2 [11] | | | DeepLabV3+ [12] | | |
|---|---|---|---|---|---|---|---|---|---|
| | 1/16 | 1/8 | 1/4 | 1/16 | 1/8 | 1/4 | 1/16 | 1/8 | 1/4 |
| ST | 65.47 | 72.24 | 75.47 | 68.45 | 72.54 | 76.21 | 73.31 | 74.20 | 77.78 |
| FST (ours) | 68.35 | 72.77 | 75.90 | 69.43 | 73.18 | 76.32 | 73.88 | 76.07 | 78.10 |
| Δ | 2.88 ↑ | 0.53 ↑ | 0.43 ↑ | 0.98 ↑ | 0.64 ↑ | 0.11 ↑ | 0.57 ↑ | 1.87 ↑ | 0.32 ↑ |

Table 6: **Evaluation on the unsupervised domain adaptation (UDA) setting** on two benchmarks. Our results are averaged over 3 random seeds.

| Method | Road | S.walk | Build. | Wall | Fence | Pole | T.light | Sign | Veget. | Terrain | Sky | Person | Rider | Car | Truck | Bus | Train | M.bike | Bike | mIoU |
|---|---|---|---|---|---|---|---|---|---|---|---|---|---|---|---|---|---|---|---|---|
| GTAV [38] → Cityscapes [13] | | | | | | | | | | | | | | | | | | | | |
| SourceOnly | 76.1 | 18.7 | 84.6 | 29.8 | 31.4 | 34.5 | 44.8 | 23.4 | 87.5 | 42.6 | 87.3 | 63.4 | 21.2 | 81.1 | 39.3 | 44.6 | 2.9 | 33.2 | 29.7 | 46.1 |
| ProDA [58] | 87.8 | 56.0 | 79.7 | 46.3 | 44.8 | 45.6 | 53.5 | 53.5 | 88.6 | 45.2 | 82.1 | 70.7 | 39.2 | 88.8 | 45.5 | 59.4 | 1.0 | 48.9 | 56.4 | 57.5 |
| CPSL [30] | 92.3 | 59.9 | 84.9 | 45.7 | 29.7 | 52.8 | 61.5 | 59.5 | 87.9 | 41.5 | 85.0 | 73.0 | 35.5 | 90.4 | 48.7 | 73.9 | 26.3 | 53.8 | 53.9 | 60.8 |
| DAFormer [24] | 95.7 | 70.2 | 89.4 | 53.5 | 48.1 | 49.6 | 55.8 | 59.4 | 89.9 | 47.9 | 92.5 | 72.2 | 44.7 | 92.3 | 74.5 | 78.2 | 65.1 | 55.9 | 61.8 | 68.3 |
| FST (ours) | 95.3 | 67.7 | 89.3 | 55.5 | 47.1 | 50.1 | 57.2 | 58.6 | 89.9 | 51.0 | 92.9 | 72.7 | 46.3 | 92.5 | 78.0 | 81.6 | 74.4 | 57.7 | 62.6 | 69.3 |
| SYNTHIA [39] → Cityscapes [13] | | | | | | | | | | | | | | | | | | | | |
| SourceOnly | 56.5 | 23.3 | 81.3 | 16.0 | 1.3 | 41.0 | 30.0 | 24.1 | 82.4 | – | 82.5 | 62.3 | 23.8 | 77.7 | – | 38.1 | – | 15.0 | 23.7 | 42.4 |
| ProDA [58] | 87.8 | 45.7 | 84.6 | 37.1 | 0.6 | 44.0 | 54.6 | 37.0 | 88.1 | – | 84.4 | 74.2 | 24.3 | 88.2 | – | 51.1 | – | 40.5 | 45.6 | 55.5 |
| CPSL [30] | 87.2 | 43.9 | 85.5 | 33.6 | 0.3 | 47.7 | 57.4 | 37.2 | 87.8 | – | 88.5 | 79.0 | 32.0 | 90.6 | – | 49.4 | – | 50.8 | 59.8 | 57.9 |
| DAFormer [24] | 84.5 | 40.7 | 88.4 | 41.5 | 6.5 | 50.0 | 55.0 | 54.6 | 86.0 | – | 89.8 | 73.2 | 48.2 | 87.2 | – | 53.2 | – | 53.9 | 61.7 | 60.9 |
| FST (ours) | 88.3 | 46.1 | 88.0 | 41.7 | 7.3 | 50.1 | 53.6 | 52.5 | 87.4 | – | 91.5 | 73.9 | 48.1 | 85.3 | – | 58.6 | – | 55.9 | 63.4 | 61.9 |

noise in pseudo-labels. Besides, as shown in Tab. 4b, the performance of longer training baselines still performs worse than our FST, which further proves the effectiveness of our method.

### 4.5 Comparison with state-of-the-art alternatives

In this subsection, we evaluate our FST with state-of-the-art approaches on the tasks of semi-supervised semantic segmentation and unsupervised domain adaptive semantic segmentation.

**Evaluation on semi-supervised segmentation.** We first evaluate the proposed FST on traditional semi-supervised semantic segmentation. As shown in Tab. 5, we compare FST (with $K = 3$) with ST on three partition protocols. Equipped with three commonly used semantic segmentation networks, *i.e.*, PSPNet [59], DeepLabV2 [11], and DeepLabV3+ [12], FST consistently improves classical ST by considerable margins. For instance, on the $1/8$ partition protocol, FST with DeepLabV3+ outperforms ST by 1.87% mIoU, showing substantial improvement. In short, FST demonstrates remarkable performance on the semi-supervised benchmark. More comprehensive comparisons between our FST against state-of-the-art alternatives can be found in *Supplementary Material*.

**Evaluation on unsupervised domain adaptive semantic segmentation.** We then compare our FST with previous self-training based competitors on UDA benchmark. The results are presented in Tab. 6. Our FST is build upon DAFormer [24], which is the state-of-the-art method currently. On GTAV → Cityscapes benchmark, our FST exceeds DAFormer by a considerable margin of 1.0% mIoU and shows dominant performance in most categories. Compared to the source only trained model, FST surpasses it by 23.2% mIoU, indicating a further step toward practical applications. The bottom part of Tab. 6 shows the comparisons on SYNTHIA → Cityscapes benchmark. FST exceeds DAFormer by 1.0% mIoU on 16 classes. Besides, our FST achieves 68.5% mIoU on 13 classes, exceeding DAFormer by 1.1%. In summary, our FST achieves new state-of-the-art performance on the UDA benchmark.

## 5 Conclusion

In this paper, we present a future-self-training framework for semantic segmentation. As an alternative to classical self-training, our approach mitigates the confirmation bias problem and achieves better performance on both UDA and semi-supervised benchmarks. The key insight of our method is to mine a model's own future states as supervision for current training. To this end, we propose two

variants, namely FST-D and FST-W, to explore the future states deeply and widely. Experiments on a wide range of settings demonstrate the effectiveness and generalizability of our methods.

**Discussion.** The major drawback of this work is that our approach is time-consuming, since we need to forward a temporary model to acquire virtual future model states. Although the number of updates to the student does *not* increase, it is better to optimize the process of obtaining future model parameters. To this end, an acceptable way is to maintain an ahead student model to provide information from future moments, which trades space for time. Besides, our approach is *general* that can be applied to other self-training frameworks such as FixMatch [41] and other tasks such as semi-supervised image recognition [44], object detection [35], few-shot learning [42], and unsupervised representation learning [20]. We will conduct further studies on these issues in the future.

**Acknowledgements.** This work was supported by the National Natural Science Foundation of China under Grants 62176017, 41871283 and U20B2069, and the Fundamental Research Funds for the Central Universities.

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
