# Learning from Future: A Novel Self-Training Framework for Semantic Segmentation – *Supplementary Material* –

**Ye Du**[1,2]    **Yujun Shen**[3]    **Haochen Wang**[4]    **Jingjing Fei**[5]    **Wei Li**[5]
**Liwei Wu**[5]    **Rui Zhao**[5,6]    **Zehua Fu**[1,2]    **Qingjie Liu**[1,2*]

[1] State Key Laboratory of Virtual Reality Technology and Systems, Beihang University
[2] Hangzhou Innovation Institute, Beihang University
[3] The Chinese University of Hong Kong
[4] Institute of Automation, Chinese Academy of Sciences    [5] SenseTime Research
[6] Qing Yuan Research Institute, Shanghai Jiao Tong University, Shanghai, China

{duyee, zehua_fu, qingjie.liu}@buaa.edu.cn    shenyujun0302@gmail.com
wanghaochen2022@ia.ac.cn    {feijingjing1, liwei1, wuliwei, zhaorui}@sensetime.com

## A  Overview

The supplementary material is organized as follows. Sec. B shows more dataset and implementation details. Sec. C provide more ablation studies of our FST, including the ablation on SYNTHIA → Cityscapes and evaluation of various segmentation decoders. Sec. D and Sec. E present more comparisons of our FST with state-of-the-art methods on both UDA and SSL benchmarks. Sec. F analyzes the training process of our method and shows more visualization comparisons with classical self-training. Sec. G discusses the social impact and potential negative impact of our work. Sec. H shows the pseudo-code of our FST.

## B  More details

**Dataset detail**. GTAV [23] contains $24,966$ labeled synthetic images with the size of $1914 \times 1052$. SYNTHIA [24] consists of $9,400$ labeled synthetic images with the size of $1280 \times 760$. Cityscapes has $2,975$ training and $500$ validation images with size of $2048 \times 1024$. PASCAL VOC 2012 [6] consists of 21 classes with $1,464$, $1,449$, and $1,456$ images for the training, validation, and test set, respectively. Following the common practice in semantic segmentation, we use the augmented training set [10] that consists of $10,582$ images for training.

**Implementation detail**. We adopt a dynamic re-weighting approach from [28] to weigh the labeled and unlabeled data, which takes the proportion of pixel-wise reliable predictions as the quality estimation of the pseudo-label:

$$\lambda = \frac{\sum_{j=1}^{H \times W} \mathbb{I}_{\max_c g_\phi(x_u)^j > \tau}}{H \times W},$$ (S1)

where $\tau$ is the confidence threshold and is set to $0.968$ for all experiments, $j$ indexes each pixel in $x_u$.

The ClassMix augmentation [28] randomly selects $1/2$ classes in the source image and paste their pixels onto the target image. The error rate of the pseudo-label is calculated by

$$\epsilon = 1 - \frac{1}{N \times C} \sum_{i=1}^{N} \sum_{c=1}^{C} \frac{\sum_{j=1}^{H \times W} \mathbb{I}_{\hat{y}_i^{j,c}=1; y_i^{j,c}=1}}{\sum_{j=1}^{H \times W} \mathbb{I}_{y_i^{j,c}=1}}.$$ (S2)

---

*Corresponding Author

36th Conference on Neural Information Processing Systems (NeurIPS 2022).

Table S1: **Comparison**. Comparison with state-of-the-art methods on SYNTHIA $\to$ Cityscapes UDA benchmark. The mIoU and the mIoU* indicate we compute mean IoU over 16 and 13 categories, respectively. The results are averaged over 3 random seeds.

| Method | Road | S.walk | Build. | Wall* | Fence* | Pole* | T.light | Sign | Veget. | Terrain | Sky | Person | Rider | Car | Truck | Bus | Train | M.bike | Bike | mIoU | mIoU* |
|---|---|---|---|---|---|---|---|---|---|---|---|---|---|---|---|---|---|---|---|---|---|
| SourceOnly | 56.5 | 23.3 | 81.3 | 16.0 | 1.3 | 41.0 | 30.0 | 24.1 | 82.4 | – | 82.5 | 62.3 | 23.8 | 77.7 | – | 38.1 | – | 15.0 | 23.7 | 42.4 | 47.7 |
| CorDA [33] | **93.3** | **61.6** | 85.3 | 19.6 | 5.1 | 37.8 | 36.6 | 42.8 | 84.9 | – | 90.4 | 69.7 | 41.8 | 85.6 | – | 38.4 | – | 32.6 | 53.9 | 55.0 | 62.8 |
| ProDA [40] | 87.8 | 45.7 | 84.6 | 37.1 | 0.6 | 44.0 | 54.6 | 37.0 | **88.1** | – | 84.4 | 74.2 | 24.3 | 88.2 | – | 51.1 | – | 40.5 | 45.6 | 55.5 | 62.0 |
| CPSL [17] | 87.2 | 43.9 | 85.5 | 33.6 | 0.3 | 47.7 | **57.4** | 37.2 | 87.8 | – | 88.5 | **79.0** | 32.0 | **90.6** | – | 49.4 | – | 50.8 | 59.8 | 57.9 | 65.3 |
| DAFormer [13] | 84.5 | 40.7 | **88.4** | 41.5 | 6.5 | 50.0 | 55.0 | **54.6** | 86.0 | – | 89.8 | 73.2 | **48.2** | 87.2 | – | 53.2 | – | 53.9 | **61.7** | 60.9 | 67.4 |
| FST (ours) | 88.3 | 46.1 | 88.0 | **41.7** | **7.3** | **50.1** | 53.6 | 52.5 | 87.4 | – | **91.5** | 73.9 | 48.1 | 85.3 | – | **58.6** | – | **55.9** | 63.4 | **61.9** | **68.5** |

Table S2: **Ablation**. Improvements on SYNTHIA $\to$ Cityscapes UDA Benchmark. Mean and SD are reported over 3 random seeds. The mIoU and the mIoU* indicate we compute mean IoU over 16 and 13 categories, respectively.

| Method | Backbone | $K$ | mIoU | $\Delta$ | mIoU* | $\Delta$* |
|---|---|---|---|---|---|---|
| ST | MiT-B5 | - | 60.9 | - | 67.4 | - |
| FST | MiT-B5 | 2 | **62.0 $\pm$ 0.9** | ↑ **1.1** | **68.8 $\pm$ 1.1** | ↑ **1.4** |
| FST | MiT-B5 | 3 | 61.9 $\pm$ 0.4 | ↑ 1.0 | 68.5 $\pm$ 0.5 | ↑ 1.1 |
| FST | MiT-B5 | 4 | 61.3 $\pm$ 1.1 | ↑ 0.4 | 68.0 $\pm$ 1.4 | ↑ 0.6 |

Following the common practice in UDA [13], we resize the images to $1024 \times 512$ pixels for Cityscapes and to $1280 \times 720$ pixels for GTAV, then a random crop of size $512 \times 512$ is used for training. ImageNet pretrained weights are used to initialize the backbones. The exception is the UPerNet with BEiT, which is initialized with the official self-supervised trained weights. The UDA models are trained on 1 Telsa A100 GPU, and the semi-supervised models are trained on 4 Telsa V100 GPUs. Our work is built on the MMSegmentation framework.

## C   More ablation

**Improvements on 13 classes.** Previous works also compare the performance on 13 classes (denoted by mIoU*), which discards three (*i.e.*, *wall*, *fence* and *pole*) of the 16 classes in SYNTHIA $\to$ Cityscapes benchmark. As shown in Tab. S1, compared with previous state-of-the-art model DAFormer, our method exceeds it by 1.1% mIoU.

**Ablation on SYNTHIA.** We also provide ablation results on SYNTHIA $\to$ Cityscapes UDA benchmark and the results are shown in Tab. S2. In the main paper, we provide experiment results with $K = 3$ to keep the same settings with the GTA $\to$ Cityscapes benchmark. However, it can be seen that $K = 2$ performs better in SYNTHIA $\to$ Cityscapes benchmark.

**Ablation on decoder.** We compare our FST with ST with various popular decoder architectures, including Atrous Spatial Pyramid Pooling (ASPP) [2], Pyramid Pooling Module (PPM) [42], PPM with Feature Pyramid Network (PPM + FPN) [35], an MLP decoder [36], and the decoder of DAFormer (SepASPP) [13]. The MLP head fuses multi-level features and upsamples the feature map to predict the segmentation mask, which is designed for Transformer-based segmentation model [36]. SepASPP is a multi-level context-aware feature fusion decoder which uses depth-wise separable convolutions to reduce over-fitting. As shown in Tab. S3, our method shows consistency improvements with these decoders.

## D   More comparisons on UDA benchmark

Most studies use CNN as the backbone. In this section, we also compare category performance of our method with other state-of-the-art CNN-based methods. As shown in Tab. S4, our FST with ResNet-101 achieves competitive performance among existing methods. Note that, we report the performances of ProDA [40] and CPSL [17] in Tab. S4 *without* knowledge distillation (which uses

Table S3: **Ablation**. Ablation on popular segmentation decoders. Experiments are done on GTA → Cityscapes benchmark. Mean and SD are reported over 3 random seeds.

| Method | Encoder | Decoder | mIoU | Δ |
|---|---|---|---|---|
| ST | ResNet-101 | MLP [36] | $55.4 \pm 1.1$ | - |
| FST | ResNet-101 | MLP [36] | $56.4 \pm 0.3$ | ↑ **1.0** |
| ST | ResNet-101 | ASPP [2] | $56.3 \pm 0.4$ | - |
| FST | ResNet-101 | ASPP [2] | $59.8 \pm 0.1$ | ↑ **3.5** |
| ST | ResNet-101 | SepASPP [13] | $56.4 \pm 0.4$ | - |
| FST | ResNet-101 | SepASPP [13] | $57.6 \pm 0.4$ | ↑ **1.2** |
| ST | ResNet-101 | PPM [42] | $56.3 \pm 0.8$ | - |
| FST | ResNet-101 | PPM [42] | $58.5 \pm 0.8$ | ↑ **2.2** |
| ST | ResNet-101 | PPM+FPN [35] | $56.6 \pm 0.9$ | - |
| FST | ResNet-101 | PPM+FPN [35] | $60.1 \pm 0.3$ | ↑ **3.5** |

Table S4: **Comparison**. Category performance comparison with state-of-the-art CNN-based methods on UDA benchmark. Methods use **ResNet-101** [11] as the backbone. The results are averaged over 3 random seeds.

| Method | Road | S.walk | Build. | Wall | Fence | Pole | T.light | Sign | Veget. | Terrain | Sky | Person | Rider | Car | Truck | Bus | Train | M.bike | Bike | mIoU |
|---|---|---|---|---|---|---|---|---|---|---|---|---|---|---|---|---|---|---|---|---|
| AdaptSeg [29] | 86.5 | 25.9 | 79.8 | 22.1 | 20.0 | 23.6 | 33.1 | 21.8 | 81.8 | 25.9 | 75.9 | 57.3 | 26.2 | 76.3 | 29.8 | 32.1 | 7.2 | 29.5 | 32.5 | 41.4 |
| ADVENT [31] | 89.4 | 33.1 | 81.0 | 26.6 | 26.8 | 27.2 | 33.5 | 24.7 | 83.9 | 36.7 | 78.8 | 58.7 | 30.5 | 84.8 | 38.5 | 44.5 | 1.7 | 31.6 | 32.4 | 45.5 |
| CBST [44] | 91.8 | 53.5 | 80.5 | 32.7 | 21.0 | 34.0 | 28.9 | 20.4 | 83.9 | 34.2 | 80.9 | 53.1 | 24.0 | 82.7 | 30.3 | 35.9 | 16.0 | 25.9 | 42.8 | 45.9 |
| PCLA [15] | 84.0 | 30.4 | 82.4 | 35.5 | 24.8 | 32.2 | 36.8 | 24.5 | 85.5 | 37.2 | 78.6 | 66.9 | 32.8 | 85.5 | 40.4 | 48.0 | 8.8 | 29.8 | 41.8 | 47.7 |
| FADA [32] | 92.5 | 47.5 | 85.1 | 37.6 | 32.8 | 33.4 | 33.8 | 18.4 | 85.3 | 37.7 | 83.5 | 63.2 | 39.7 | 87.5 | 32.9 | 47.8 | 1.6 | 34.9 | 39.5 | 49.2 |
| MCS [4] | 92.6 | 54.0 | 85.4 | 35.0 | 26.0 | 32.4 | 41.2 | 29.7 | 85.1 | 40.9 | 85.4 | 62.6 | 34.7 | 85.7 | 35.6 | 50.8 | 2.4 | 31.0 | 34.0 | 49.7 |
| CAG [41] | 90.4 | 51.6 | 83.8 | 34.2 | 27.8 | 38.4 | 25.3 | 48.4 | 85.4 | 38.2 | 78.1 | 58.6 | 34.6 | 84.7 | 21.9 | 42.7 | 41.1 | 29.3 | 37.2 | 50.2 |
| FDA [39] | 92.5 | 53.3 | 82.4 | 26.5 | 27.6 | 36.4 | 40.6 | 38.9 | 82.3 | 39.8 | 78.0 | 62.6 | 34.4 | 84.9 | 34.1 | 53.1 | 16.9 | 27.7 | 46.4 | 50.5 |
| IAST [21] | 93.8 | 57.8 | 85.1 | 39.5 | 26.7 | 26.2 | 43.1 | 34.7 | 84.9 | 32.9 | 88.0 | 62.6 | 29.0 | 87.3 | 39.2 | 49.6 | 23.2 | 34.7 | 39.6 | 51.5 |
| DACS [28] | 89.9 | 39.7 | 87.9 | 30.7 | 39.5 | 38.5 | 46.4 | 52.8 | 88.0 | 44.0 | 88.8 | 67.2 | 35.8 | 84.5 | 45.7 | 50.2 | 0.0 | 27.3 | 34.0 | 52.1 |
| RCCR [43] | 93.7 | 60.4 | 86.5 | 41.0 | 32.0 | 37.3 | 38.7 | 38.6 | 87.2 | 43.0 | 85.5 | 65.4 | 35.1 | 88.3 | 41.8 | 51.6 | 0.0 | 38.0 | 52.1 | 53.5 |
| MetaCo [9] | 92.8 | 58.1 | 86.2 | 39.7 | 33.1 | 36.3 | 42.0 | 38.6 | 85.5 | 37.8 | 87.6 | 62.8 | 31.7 | 84.8 | 35.7 | 50.3 | 2.0 | 36.8 | 48.0 | 52.1 |
| CTF [20] | 92.5 | 58.3 | 86.5 | 27.4 | 28.8 | 38.1 | 46.7 | 42.5 | 85.4 | 38.4 | 91.8 | 66.4 | 37.0 | 87.8 | 40.7 | 52.4 | 44.6 | 41.7 | 59.0 | 56.1 |
| CorDA [33] | 94.7 | 63.1 | 87.6 | 30.7 | 40.6 | 40.2 | 47.8 | 51.6 | 87.6 | 47.0 | 89.7 | 66.7 | 35.9 | 90.2 | 48.9 | 57.5 | 0.0 | 39.8 | 56.0 | 56.6 |
| ProDA [40] | 91.5 | 52.4 | 82.9 | 42.0 | 35.7 | 40.0 | 44.4 | 43.3 | 87.0 | 43.8 | 79.5 | 66.5 | 31.4 | 86.7 | 41.1 | 52.5 | 0.0 | 45.4 | 53.8 | 53.7 |
| CPSL [17] | 91.7 | 52.9 | 83.6 | **43.0** | 32.3 | **43.7** | **51.3** | 42.8 | 85.4 | 37.6 | 81.1 | 69.5 | 30.0 | 88.1 | 44.1 | **59.9** | 24.9 | **47.2** | 48.4 | 55.7 |
| FST (ours) | **95.0** | **65.1** | **88.4** | 40.1 | 36.8 | 38.0 | 50.2 | **55.9** | **88.1** | 45.8 | 88.7 | **70.1** | **45.0** | 87.4 | 45.3 | 54.8 | 37.2 | 45.6 | 58.9 | **59.8** |

self-supervised trained models) for a fair comparison. On the SYNTHIA → Cityscapes benchmark, we set $\mu' = 0.9999$ for our FST. As shown in Tab. S5, our method also demonstrates competitive performance, which is slightly lower than CPSL, a class-balanced training approach that is orthogonal to our work.

# E   More comparisons on SSL benchmark

We compare our FST with previous state-of-the-art semi-supervised semantic segmentation frameworks, including CCT [22], GCT [16] and CPS [3]. These frameworks do *not* use CutMix Augmentation [7] for fair comparisons. Experiments are conducted on both the PASCAL VOC 2012 and Cityscapes, with $1/16$, $1/8$ and $1/4$ samples as the labeled data. The comparisons are shown in Tab. S6. Note that some works such as AEL [14] are not included here, since we compare our FST with the basic SSL frameworks. However, AEL focuses on the long tail problem under a ST framework, which is orthogonal to our work. On PASCAL VOC 2012, our FST achieves the best performance among these SSL frameworks. On Cityscapes, our method exceeds CCT and GCT by large margins. Compared to CPS, our FST also achieves competitive results. Our FST uses minimal data augmentations, thus its performance could be further boosted by advanced augmentation strategies. These results show the effectiveness of the proposed FST on the traditional SSL benchmark.

Table S5: **Comparison**. Comparison with state-of-the-art CNN-based methods on SYNTHIA → Cityscapes UDA benchmark. Methods use **ResNet-101** [11] as the backbone. The results are averaged over 3 random seeds. The mIoU and the mIoU* are calculated over 16 and 13 categories, respectively.

| Method | Road | S.walk | Build. | Wall* | Fence* | Pole* | T.light | Sign | Veget. | Terrain | Sky | Person | Rider | Car | Truck | Bus | Train | M.bike | Bike | mIoU | mIoU* |
|---|---|---|---|---|---|---|---|---|---|---|---|---|---|---|---|---|---|---|---|---|---|
| AdaptSeg [29] | 79.2 | 37.2 | 78.8 | – | – | – | 9.9 | 10.5 | 78.2 | – | 80.5 | 53.5 | 19.6 | 67.0 | – | 29.5 | – | 21.6 | 31.3 | – | 45.9 |
| ADVENT [31] | 85.6 | 42.2 | 79.7 | 8.7 | 0.4 | 25.9 | 5.4 | 8.1 | 80.4 | – | 84.1 | 57.9 | 23.8 | 73.3 | – | 36.4 | – | 14.2 | 33.0 | 41.2 | 48.0 |
| CBST [44] | 68.0 | 29.9 | 76.3 | 10.8 | 1.4 | 33.9 | 22.8 | 29.5 | 77.6 | – | 78.3 | 60.6 | 28.3 | 81.6 | – | 23.5 | – | 18.8 | 39.8 | 42.6 | 48.9 |
| FDA [39] | 79.3 | 35.0 | 73.2 | – | – | – | 19.9 | 24.0 | 61.7 | – | 82.6 | 61.4 | 31.1 | 83.9 | – | 40.8 | – | 38.4 | 51.1 | – | 52.5 |
| FADA [32] | 84.5 | 40.1 | 83.1 | **4.8** | 0.0 | 34.3 | 20.1 | 27.2 | 84.8 | – | 84.0 | 53.5 | 22.6 | 85.4 | – | 43.7 | – | 26.8 | 27.8 | 45.2 | 52.5 |
| MCS [4] | 88.3 | **47.3** | 80.1 | – | – | – | 21.6 | 20.2 | 79.6 | – | 82.1 | 59.0 | 28.2 | 82.0 | – | 39.2 | – | 17.3 | 46.7 | – | 53.2 |
| PyCDA [18] | 75.5 | 30.9 | 83.3 | 20.8 | 0.7 | 32.7 | 27.3 | 33.5 | 84.7 | – | 85.0 | 64.1 | 25.4 | 85.0 | – | 45.2 | – | 21.2 | 32.0 | 46.7 | 53.3 |
| PLCA [15] | 82.6 | 29.0 | 81.0 | 11.2 | 0.2 | 33.6 | 24.9 | 18.3 | 82.8 | – | 82.3 | 62.1 | 26.5 | 85.6 | – | 48.9 | – | 26.8 | 52.2 | 46.8 | 54.0 |
| RCCR [43] | 79.4 | 45.3 | 83.3 | – | – | – | 24.7 | 29.6 | 68.9 | – | 87.5 | 61.1 | 33.8 | 87.0 | – | 51.0 | – | 32.1 | 52.1 | – | 56.8 |
| IAST [21] | 81.9 | 41.5 | 83.3 | 17.7 | **4.6** | 32.3 | 30.9 | 28.8 | 83.4 | – | 85.0 | 65.5 | 30.8 | 86.5 | – | 38.2 | – | 33.1 | 52.7 | 49.8 | 57.0 |
| SAC [1] | **89.3** | 47.2 | **85.5** | 26.5 | 1.3 | **43.0** | 45.5 | 32.0 | **87.1** | – | **89.3** | 63.6 | 25.4 | 86.9 | – | 35.6 | – | 30.4 | 53.0 | 52.6 | 59.3 |
| ProDA [40] | 87.1 | 44.0 | 83.2 | **26.9** | 0.7 | 42.0 | 45.8 | 34.2 | **86.7** | – | 81.3 | 68.4 | 22.1 | 87.7 | – | 50.0 | – | 31.4 | 38.6 | 51.9 | 58.5 |
| CPSL [17] | 87.3 | 44.4 | 83.8 | 25.0 | 0.4 | 42.9 | **47.5** | 32.4 | 86.5 | – | 83.3 | **69.6** | 29.1 | **89.4** | – | **52.1** | – | 42.6 | 54.1 | **54.4** | **61.7** |
| FST (ours) | 68.5 | 28.9 | **85.5** | 21.1 | 3.3 | 40.4 | 46.3 | **53.0** | 77.6 | – | 85.3 | 69.5 | **42.4** | 87.0 | – | 48.5 | – | **46.4** | **60.0** | 54.0 | 61.5 |

Table S6: **Comparison**. Comparison with state-of-the-art semi-supervised semantic segmentation methods on the validation set. We use FST-D with $K = 3$ and † means results reported by [34].

| Method | 1/16 | 1/8 | 1/4 |
|---|---|---|---|
| SupOnly† | 67.87 | 71.55 | 75.80 |
| CutMix† [7] | 71.66 | 75.51 | 77.33 |
| CCT [22] | 71.86 | 73.68 | 76.51 |
| GCT [16] | 70.90 | 73.29 | 76.66 |
| CPS [3] | 72.18 | 75.83 | 77.55 |
| FST (ours) | **73.88** | **76.07** | **78.10** |

(a) PASCAL VOC 2012 [6].

| Method | 1/16 | 1/8 | 1/4 |
|---|---|---|---|
| SupOnly† | 65.74 | 72.53 | 74.43 |
| CutMix† [7] | 67.06 | 71.83 | 76.36 |
| CCT [22] | 69.32 | 74.12 | 75.99 |
| GCT [16] | 66.75 | 72.66 | 76.11 |
| CPS [3] | 70.50 | **75.71** | **77.41** |
| FST (ours) | **71.03** | 75.36 | 76.61 |

(b) Cityscapes [5].

# F   More analyses

Fig. S1 presents more performance (mIoU) curves of various network architectures. We calculate mIoU on the validation set every 2,000 iterations and plot the mean and standard deviation over 3 random seeds. During training, our FST quickly achieves the performance of classical ST, benefiting from the guidance of the estimated future model states. Moreover, to verify the effect on reducing the confirmation bias, we further observe the training loss on the labeled data (*i.e.*, the training data of source domain), which serves as a complementary to Fig. S1. The confirmation bias is considered to mislead the model training. Here, inspired by [19], we empirically observe the bias issue through the model's own training error on the labeled data, since a biased model struggles to fit the labeled samples. As shown in Fig. S2, our FST shows lower cross-entropy loss value of each iteration, especially in the early training stages. This phenomenon further proves that our FST indeed mitigates the bias problem to some extent. Note that the presented value in the figure maintains an EMA of the CE loss during training and we plot the mean and standard deviation over 3 random seeds. As a comparison, we also plot the training error on the unlabeled data of each iteration, which is shown in Fig. S3. Our FST generates higher-quality pseudo-labels on unlabeled samples and achieves lower training error on these samples. On the one hand, better pseudo-labels make the learning process easier. On the other hand, due to the mitigation of the confirmation bias, the model reduces the over-fitting to noise pseudo-labels. Finally, we show more visualization results in Fig. S4 for more qualitative comparisons between ST and our FST.

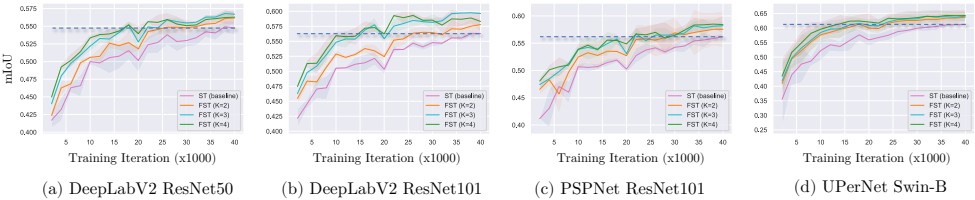

Figure S1: **Analyses.** Performance curve on validation set during training.

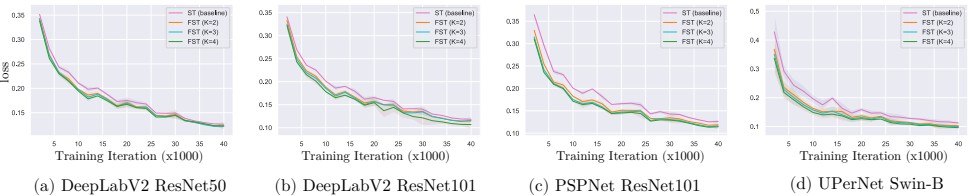

Figure S2: **Analyses.** Cross-entropy loss on the labeled (training) data during training.

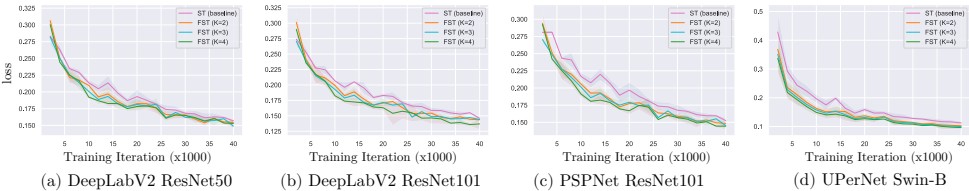

Figure S3: **Analyses.** Cross-entropy loss on the unlabeled (training) data during training.

## G  More discussion

**Broader impact.** This work mainly focuses on semantic segmentation and its widely adopted momentum teacher-based self-training framework. However, our approach is a general framework that could be applied to other tasks such as image recognition [27], object detection [38], few-shot learning [26] and unsupervised representation learning [8]. When it comes to other popular online self-training frameworks such as FixMatch [25], Noisy student [37] and Cycle self-training [19], our method is easy to extend by modifying the way of exploiting a model's own future model states. Besides, our work is compatible with existing appealing technologies such as contrastive learning [12] and active learning [30]. We hope our approach can inspire further research about new algorithms, theoretical analyses and applications.

**Potential negative impact.** Our work improves the utilization of unlabeled data for semantic segmentation, which could benefit many useful applications such as autonomous driving and remote sensing image analysis. However, this technology may also be applied to some controversial applications such as surveillance. This is a potential risk and a common problem of existing deep learning algorithms and is gaining public attention. Another possible negative impact is that the learned model could be biased if there was bias in the training data. Besides, the corresponding carbon emission problem should be considered due to the large-scale data and long-time training of our work.

## H  Pseudo-code

To makes our FST easy to understand, we provide pseudo-code in a Pytorch-like style. To simplify, the improved version of FST (*i.e.*, Eq. (4)) is implemented in Alg. S1.

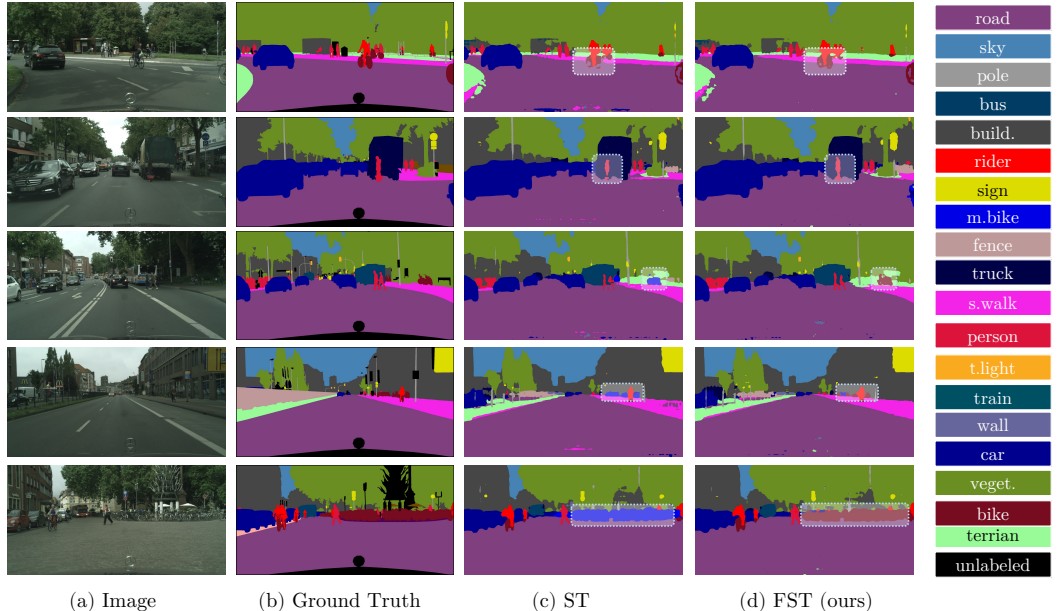

(a) Image      (b) Ground Truth      (c) ST      (d) FST (ours)

Figure S4: **Analyses.** More qualitative results on Cityscapes validation set. DeepLabV2 [2] with ResNet-101 [11] is used.

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

**Algorithm S1** Pseudo-code of FST in a PyTorch-like style.

```
1  # g_s , g_t: the student model and the teacher model
2  # mu , mu': momentum for EMA
3  # Lambda: dynamic weight to balance the labeled and unlabeled data
4
5  g_t.params = g_s.params  # initialize
6
7  for (x_l, y_l), x_u in loader:  # load samples
8      # momentum update with previous student states
9      g_t.params = mu*g_t.params+(1-mu)*g_s.params
10     # cache the current student
11     g_tmp = g_s.copy()
12     # pseudo label prediction: for temp network
13     with no_grad():
14         y_u = argmax(g_t.forward(x_u))
15
16     # train the temp model
17     loss_l = CrossEntropyLoss(g_tmp.forward(x_l), y_l)
18     loss_u = CrossEntropyLoss(g_tmp.forward(x_u), y_u)
19     loss_virtual = loss_l + Lambda * loss_u   # calculate the loss for temp model
20
21     loss_virtual.backward()
22     update(g_tmp.params)     # SGD update: temp network
23
24     # momentum update with future student states
25     g_t.params = mu_prime * g_t.params + (1-mu_prime) * g_tmp.params
26     # pseudo label prediction: for student network
27     with no_grad():
28         y_u = argmax(g_t.forward(x_u))
29
30     # train the student
31     loss_l = CrossEntropyLoss(g_s.forward(x_l), y_l)
32     loss_u = CrossEntropyLoss(g_s.forward(x_u), y_u)
33     loss = loss_l + Lambda * loss_u   # calculate loss for student model
34
35     loss.backward()
36     update(g_s.params)   # SGD update: student network
37
38     # delete cache
39     del(g_tmp)
```

[13] L. Hoyer, D. Dai, and L. Van Gool. Daformer: Improving network architectures and training strategies for domain-adaptive semantic segmentation. In *IEEE Conf. Comput. Vis. Pattern Recog.*, 2021.

[14] H. Hu, F. Wei, H. Hu, Q. Ye, J. Cui, and L. Wang. Semi-supervised semantic segmentation via adaptive equalization learning. *Adv. Neural Inform. Process. Syst.*, 2021.

[15] G. Kang, Y. Wei, Y. Yang, Y. Zhuang, and A. Hauptmann. Pixel-level cycle association: A new perspective for domain adaptive semantic segmentation. In *Adv. Neural Inform. Process. Syst.*, 2020.

[16] Z. Ke, D. Qiu, K. Li, Q. Yan, and R. W. Lau. Guided collaborative training for pixel-wise semi-supervised learning. In *Eur. Conf. Comput. Vis.*, 2020.

[17] R. Li, S. Li, C. He, Y. Zhang, X. Jia, and L. Zhang. Class-balanced pixel-level self-labeling for domain adaptive semantic segmentation. In *IEEE Conf. Comput. Vis. Pattern Recog.*, 2022.

[18] Q. Lian, F. Lv, L. Duan, and B. Gong. Constructing self-motivated pyramid curriculums for cross-domain semantic segmentation: A non-adversarial approach. In *Int. Conf. Comput. Vis.*, 2019.

[19] H. Liu, J. Wang, and M. Long. Cycle self-training for domain adaptation. *Adv. Neural Inform. Process. Syst.*, 2021.

[20] H. Ma, X. Lin, Z. Wu, and Y. Yu. Coarse-to-fine domain adaptive semantic segmentation with photometric alignment and category-center regularization. In *IEEE Conf. Comput. Vis. Pattern Recog.*, 2021.

[21] K. Mei, C. Zhu, J. Zou, and S. Zhang. Instance adaptive self-training for unsupervised domain adaptation. In *Eur. Conf. Comput. Vis.*, 2020.

[22] Y. Ouali, C. Hudelot, and M. Tami. Semi-supervised semantic segmentation with cross-consistency training. In *IEEE Conf. Comput. Vis. Pattern Recog.*, 2020.

[23] S. R. Richter, V. Vineet, S. Roth, and V. Koltun. Playing for data: Ground truth from computer games. In *Eur. Conf. Comput. Vis.*, 2016.

[24] G. Ros, L. Sellart, J. Materzynska, D. Vazquez, and A. M. Lopez. The synthia dataset: A large collection of synthetic images for semantic segmentation of urban scenes. In *IEEE Conf. Comput. Vis. Pattern Recog.*, 2016.

[25] K. Sohn, D. Berthelot, N. Carlini, Z. Zhang, H. Zhang, C. A. Raffel, E. D. Cubuk, A. Kurakin, and C.-L. Li. Fixmatch: Simplifying semi-supervised learning with consistency and confidence. *Adv. Neural Inform. Process. Syst.*, 2020.

[26] J.-C. Su, S. Maji, and B. Hariharan. When does self-supervision improve few-shot learning? In *Eur. Conf. Comput. Vis.*, 2020.

[27] A. Tarvainen and H. Valpola. Mean teachers are better role models: Weight-averaged consistency targets improve semi-supervised deep learning results. *Adv. Neural Inform. Process. Syst.*, 2017.

[28] W. Tranheden, V. Olsson, J. Pinto, and L. Svensson. Dacs: Domain adaptation via cross-domain mixed sampling. In *IEEE Winter Conf. Appl. Comput. Vis.*, 2021.

[29] Y.-H. Tsai, W.-C. Hung, S. Schulter, K. Sohn, M.-H. Yang, and M. Chandraker. Learning to adapt structured output space for semantic segmentation. In *IEEE Conf. Comput. Vis. Pattern Recog.*, 2018.

[30] A. Vezhnevets, J. M. Buhmann, and V. Ferrari. Active learning for semantic segmentation with expected change. In *IEEE Conf. Comput. Vis. Pattern Recog.*, 2012.

[31] T.-H. Vu, H. Jain, M. Bucher, M. Cord, and P. Pérez. Advent: Adversarial entropy minimization for domain adaptation in semantic segmentation. In *IEEE Conf. Comput. Vis. Pattern Recog.*, 2019.

[32] H. Wang, T. Shen, W. Zhang, L.-Y. Duan, and T. Mei. Classes matter: A fine-grained adversarial approach to cross-domain semantic segmentation. In *Eur. Conf. Comput. Vis.*, 2020.

[33] Q. Wang, D. Dai, L. Hoyer, L. Van Gool, and O. Fink. Domain adaptive semantic segmentation with self-supervised depth estimation. In *Int. Conf. Comput. Vis.*, 2021.

[34] Y. Wang, H. Wang, Y. Shen, J. Fei, W. Li, G. Jin, L. Wu, R. Zhao, and X. Le. Semi-supervised semantic segmentation using unreliable pseudo-labels. *IEEE Conf. Comput. Vis. Pattern Recog.*, 2022.

[35] T. Xiao, Y. Liu, B. Zhou, Y. Jiang, and J. Sun. Unified perceptual parsing for scene understanding. In *Eur. Conf. Comput. Vis.*, 2018.

[36] E. Xie, W. Wang, Z. Yu, A. Anandkumar, J. M. Alvarez, and P. Luo. Segformer: Simple and efficient design for semantic segmentation with transformers. *Adv. Neural Inform. Process. Syst.*, 2021.

[37] Q. Xie, M.-T. Luong, E. Hovy, and Q. V. Le. Self-training with noisy student improves imagenet classification. In *IEEE Conf. Comput. Vis. Pattern Recog.*, 2020.

[38] M. Xu, Z. Zhang, H. Hu, J. Wang, L. Wang, F. Wei, X. Bai, and Z. Liu. End-to-end semi-supervised object detection with soft teacher. In *Int. Conf. Comput. Vis.*, 2021.

[39] Y. Yang and S. Soatto. Fda: Fourier domain adaptation for semantic segmentation. In *IEEE Conf. Comput. Vis. Pattern Recog.*, 2020.

[40] P. Zhang, B. Zhang, T. Zhang, D. Chen, Y. Wang, and F. Wen. Prototypical pseudo label denoising and target structure learning for domain adaptive semantic segmentation. In *IEEE Conf. Comput. Vis. Pattern Recog.*, 2021.

[41] Q. Zhang, J. Zhang, W. Liu, and D. Tao. Category anchor-guided unsupervised domain adaptation for semantic segmentation. *Adv. Neural Inform. Process. Syst.*, 2019.

[42] H. Zhao, J. Shi, X. Qi, X. Wang, and J. Jia. Pyramid scene parsing network. In *IEEE Conf. Comput. Vis. Pattern Recog.*, 2017.

[43] Q. Zhou, C. Zhuang, X. Lu, and L. Ma. Domain adaptive semantic segmentation with regional contrastive consistency regularization. In *Int. Conf. Multimedia and Expo*, 2022.

[44] Y. Zou, Z. Yu, B. Kumar, and J. Wang. Unsupervised domain adaptation for semantic segmentation via class-balanced self-training. In *Eur. Conf. Comput. Vis.*, 2018.