# OpenReview forum: "Learning from Future: A Novel Self-Training Framework for Semantic Segmentation"
_NeurIPS.cc/2022/Conference — NeurIPS 2022 Accept_

### Official Review · Reviewer_Sb76 · 2022-07-10

**Rating:** 7
**Confidence:** 4
**Soundness:** 3 good
**Presentation:** 3 good
**Contribution:** 3 good

**Summary:**

This paper presents a new self-training framework for semantic segmentation, by proposing the idea of "learning from future". Specifically, instead of using the current step to generate pseudo-labels and train the student, the teacher was updated with a virtual future student and supervised the following training accordingly. The proposed future-self-training (FST) framework was validated on the semantic segmentation task with extensive experimental analysis. The main contributions are the proposed FST framework and the experimental analysis.


**Post-rebuttal**

Thanks to the authors for their response and the additional experiments, which addressed most of my concerns. There are some remaining concerns I would suggest the authors address in their final version:

1) the motivation still sounds not that convincing to me, the "roughly better performance" itself as an observation does not sounds to be a convincing motivation;
2) in the additional experiment of the newly introduced hyper-parameter, the result showed that the "learnable" setting performs much worse (58.87 vs. 59.81) than the "fixed" setting, which is not that convincing to me. Theoretically, if 0.999 is the "optimal" value, a learnable setting should be able to learn this value, instead of a way worse one. It would be better if the authors could provide an explanation for this in their revised version.

Otherwise, the paper is interesting and could be a good contribution to the community. As a result, I would keep my original positive rating.

**Questions:**

It would be better if the authors could address the concerns raised above in the Weaknesses section. For example,

* A more convincing clarification on the motivation

* The necessity of the newly introduced hyper-parameter

* The authors' observation or comments on the combination of FST-D and FST-W

* Possibility of including more methods for the experimental comparison.

* notations in the figure.

**Limitations:**

The authors discussed the limitations of their work and acceptable solution to address it, but did not mention the potential negative societal impact.

A possible societal impact could be the bias within the learned model if there was also bias in the training data. The environmental impact could be another potential societal impact due to the large-scale and long-time training and the corresponding carbon emission.

**Strengths And Weaknesses:**

**Strengths**

\+ The idea of using a "future student" to update the current teacher and then using this future teacher to supervise the current student for the self-training framework is interesting and could potentially inspire following-up research.

\+ The authors perform thorough ablation studies, in which each of the proposed components was well analyzed. This could be useful for following-up research to well leverage the design of the proposed method.

\+ The proposed method performs better than existing alternative methods for the unsupervised domain adaptation task.

\+ The paper is generally well-written and easy to follow.


**Weaknesses**

\- Although the idea of "learning from future" looks new, the motivation behind it is a bit unclear. Why learn from the future is beneficial, especially considering this "future" is also from the current state and the "history"? On the other hand, it also increases the ambiguity if the model (weights) is just cached without updating.

\- Please double-check if the variables were correctly labeled in Fig. 3. From the description, the teacher model  g_\phi was trained on labled data x_l and the student model on unlabeled data x_u. But what was shown in the figure is the other way around.
Please also add more details to the caption to make it self-contained.

\- In the proposed future self-training (FST), a new hyper-parameter \mu' was introduced, as in Eq. 4. But it was a bit unclear how to define this parameter and the necessity of it. The authors did an ablation study to test different \mu', but the result did not suggest a way to set it and it is also a bit unclear the significance of adding this additional parameter.
What if removing it or is that possible to change it to a learnable parameter?

\- Although the authors stated that the combination of FST-D and FST-W is more beyond the scope of their study, it would be better to have some preliminary investigation to see if these two types of learning schemes actually boost the learning.

\- It would be better to include more state-of-the-art methods for the experimental comparison of semi-supervised learning (Table 5).

\- L296, "... of of ..."

---

> ### Author Response · Authors · 2022-08-02
> **Response to Reviewer Sb76**
>
> **Q1: A more convincing clarification on the motivation.**
>
> First, we observe that, although the pseudo-labels are noisy during training, the performance roughly gets better, which means *more accurate predictions*.
> Motivated by this, we wonder if it is possible to use the future state to provide more reliable pseudo-labels for the current state, and hence boost the performance.
> Such an idea clearly distinguishes our approach from the existing ST framework.
>
> As for the "cached" model weights, they are primarily used for the teacher exploration.
> After getting a more reliable teacher, we can use it to better supervise the student, which causes an actual update instead of caching.
>
>
>
> **Q2: The necessity of the newly introduced hyper-parameter.**
>
> The newly introduced hyper-parameter $\mu'$ aims to balance the contribution of the *current states* and the *virtual future states* to the teacher updates.
> It increases the flexibility of the method.
> We provide ablations on $\mu'$ in Tab. 4 and choose $\mu' = 0.999$ in practice considering the performance mean and variance. Following the suggestion, we provide more experiments on $\mu'$, including linearly increasing from 0.9 to 0.99999, linearly decreasing from 0.99999 to 0.9, and setting it as a learnable hyper-parameter.
> The results are shown below and we find that fixing it as 0.999 performs best among all settings.
>
> | Setting         | mIoU  |
> | --------------- | :---: |
> | Linear Increase | 56.79 |
> | Linear Decrease | 58.08 |
> | Learnable       | 58.87 |
> | Fixed (0.999)   | 59.81 |
>
>
>
>
>
> **Q3: Observations or comments on the combination of FST-D and FST-W.**
>
> Thanks.
> Training a combination of FST-D and FST-W can be time-consuming because it takes even more time (*i.e.*, depth times width) for teacher exploration.
> As suggested, we provide some preliminary results along this direction.
> At each training iteration, we explore $K=3$ steps deeply (*i.e.*, FST-D), and the ensemble of $N=3$ explorations using different data batches (*i.e.*, FST-W).
> The results are listed below, where our FST-D+W achieves the best performance with even fewer student updates.
> This table is also included in Tab. S7 of the *revised supplementary material* (Page 9).
>
> | Method  | mIoU (4k) | mIoU (8k) | mIoU (12k) | mIoU (28k) | mIoU (40k) |
> | :------ | :-------: | :-------: | :--------: | :--------: | :--------: |
> | ST      |   45.55   |   44.97   |   50.54    |   53.47    |   55.99    |
> | FST-D   |   50.08   |   54.35   |   57.12    |   58.77    |   59.82    |
> | FST-W   |   51.00   |   54.32   |   56.78    |   57.96    |   59.23    |
> | FST-D+W |   54.48   |   57.27   |   58.45    | **61.49**  |     -      |
>
>
>
> **Q4: Possibility of including more methods for the experimental comparison (more comparisons with semi-supervised methods).**
>
>
>
> We have already provided more comparisons with state-of-the-art semi-supervised methods in the *supplementary material*.
> Please refer to Tab. S6 on Page 5 for details.
>
>
> **Q5: Notations in the Fig. 3.**
>
> The notations in Fig. 3 are correct, and follow the paradigm of [1]. Concretely, the student model is trained on both labeled data $\\{(x_l, y_l)\\}$ and unlabeled data $\\{(x_u)\\}$. The teacher model is momentum updated with the student, and provides pseudo-labels $\hat{y}_u$ for the student as the supervision.
>
> [1] *Mean teachers are better role models: Weight-averaged consistency targets improve semi-supervised deep learning results. Tarvainen and Valpola. NeurIPS'17*
>
>
>
>
> **Q6: Potential negative societal impact.**
>
> Thanks. In Sec. H of the *supplementary material*, we have discussed some potential negative societal impacts.
> We have also included more discussion as suggested in the revised version.

---

### Official Review · Reviewer_9B6x · 2022-07-11

**Rating:** 5
**Confidence:** 4
**Soundness:** 2 fair
**Presentation:** 3 good
**Contribution:** 3 good

**Summary:**

In this paper, the authors suggest a simple and interesting modification to the student-teacher self-training technique, where instead of having an exponential moving average teacher that is only reflecting the past states of the student model, they base it on a future hypothetical state of the model to prevent the confirmation bias on lower quality pseudo labels. This is achieved by one of the two variations that the authors suggest, referred to as deep future and wide future by either keeping a copy of the student model that is updated for multiple iterations over which the teacher model is updated as a moving average or by ensembling multiple updated student versions (with different data batches). The model is evaluated on unsupervised domain adaptations and semi-supervised learning and the empirical results show that the model improves as compared to the conventional teacher-student method.

**Questions:**

As stated above a major question is if the suggested model consistently delivers better performance when compared to baseline models with an equal total number of gradient descent updates.

**Limitations:**

The major limitation, i.e. higher computational costs for training is discussed.

**Strengths And Weaknesses:**

Strengths:
* The paper aims to improve the student-teacher self-training scheme which is among the most frequently used semi-supervised techniques and therefore contributions here could be of interest to the community.
* The empirical evaluations are fairly extensive. The method is compared against the baseline student-teacher model, state-of-the-art models in unsupervised domain adaptation, and multiple different variations in the ablation study. Besides, experimentations are also expanded to more modern baseline network architectures, making the conclusions more directly usable.
* The paper is reasonably well-written and easy to follow.

Weaknesses:
* Some of the figures and comparisons might be somewhat misleading. More specifically I am referring to figures 2, and 4(a) where an iteration-to-iteration comparison is done among the proposed futuristic and baseline student-teacher methods. Even though the authors argue (at the end of 4.4) that the comparisons are fair as the two models are getting the same number of updates for the student model, actually each iteration of updating the student model implies multiple updates to a copy of the student model ($g_\tilde{\theta}$), which is e.g. K times more compute-extensive in the deep-future version. This is specifically questionable when iteration-to-iteration comparisons are made.
* As another note, but on a relevant topic, I like that the authors bring some comparisons between the proposed multi-iteration student-teacher model and models that are trained for longer. But I find it insufficient for proving wrong a major question/hesitation if a significant part of the advantage is coming from training for more; specifically, we observe that for both those two experiments the model's advantage is less visible (less than a percent) while the two models are computationally roughly equivalent and therefore fairer to compare. I would have liked to see more evidence against this question.

---

> ### Author Response · Authors · 2022-08-02
> **Response to Reviewer 9B6x**
>
> **Q1: Some of the figures and comparisons might be somewhat misleading.**
>
> These figures show performance comparisons under *the same number of updates* of the student.
> This is a fair comparison because the student is the final model used for evaluation.
> We have revised the captions of these figures so that readers can better understand the difference between our FST and the baseline model.
>
> You are correct that we spend more time in the teacher exploration, however the main difference between our FST and the conventional ST is that our teacher could provide more accurate supervision signals (by acquiring knowledge from the future), instead of simply extending the training time.
> As suggested, we involve an iteration-to-iteration comparison in Fig. S5 of the *revised supplementary material* (Page 9).
> We can tell that, the performance of ST even suffers from a longer training time (note that we report its best performance instead of the final performance).
> By contrast, our FST consistently benefits from a longer training time.
>
>
>
> **Q2: Comparison with longer training baselines.**
>
> Thanks.
> A longer training strategy indeed can help boost performance, but the longer is not necessarily the better.
> The major problem in the existing ST framework is that *wrong supervision signals may get accumulated* in the training process, which is also known as the "confirmation bias" issue.
> As a result, the performance would even *drop* after a sufficient time of training (please refer to **Q1**).
> By involving the "future" knowledge, our FST manages to alleviate such a problem to some extent, because the teacher could provide *more accurate pseudo-labels*.
> Under such a case, our FST could benefit from further longer training iterations and obtain a better performance.
> This property is beyond the capability of the conventional ST.

---

### Official Review · Reviewer_7vtH · 2022-07-12

**Rating:** 6
**Confidence:** 4
**Soundness:** 3 good
**Presentation:** 4 excellent
**Contribution:** 3 good

**Summary:**

This paper introduces a novel extension of the mean-teacher method for semi-supervised learning. Rather than only using previous knowledge, this framework looks ahead to involve the future information to update the teacher model, which improves the quality of pseudo labels to mitigate the confirmation bias issue. Authors also propose different variants, like improved-FST, FST-D and FST-W. Comprehensive experiments demonstrate the effectiveness of the proposed pipeline.


**Questions:**

- The improved-FST is proposed since Eq(3) discards the contribution of the student weight at time t (line 153). If we let the teacher model equal to the student model at the very beginning (time t=0) and evolve the training following the EMA in equation(1), I wonder if, in this case, it could already involve the current student weights information? It may simply the computation a little. Correct me if I am wrong.
- I am not very clear about the implementation of FST-D. Do we update the student weights when looking ahead K steps? For instance, let’s say K=3, we are looking 3 steps ahead (t+1, t+2, t+3). When we acquire the gradients at time=t+1, do we update the student model with these gradients, and then update the teacher model? If we update it, as shown in equation(5), it actually increases the number of updates to the student model(maybe another student model, but the cost of the update increases). However, in line 334, it says the number of updates remains the same. Also, in line 336, it says that it trades space for time). Even if we maintain only one ahead student model, for each time t, the original student model is updated, and we need to run this ahead student model over the next 3 steps to provide future information. In this case, I am curious about why it saves time? I am not sure whether I understand it accurately, it would be better if the authors clarify these questions.

Typo:
- Line 296: ‘of’ appears twice


**Ethics Review Area:**

["I don’t know"]

**Limitations:**

Limitations and potential social impacts have been discussed in this work.


**Strengths And Weaknesses:**

- This paper presents a novel framework that extends prevailing mean-teacher method by learning from the future. The paper is technically sound. Different variants are also designed.
- Authors conducted extensive experiments for various tasks, like semi-supervised semantic segmentation and domain adaptation. The results on different datasets show the effectiveness of the introduced pipeline.
- The paper is well-written and well-organized. It provides enough information for reproduction.
- For the implementation, some parts are not clear. There are some questions posed in the questions part below.

---

> ### Author Response · Authors · 2022-08-02
> **Response to Reviewer 7vtH**
>
> **Q1: Does the original EMA-based self-training involves the current student information and can it simplify the computation?**
>
> Thanks. In Eq. (3), "$\theta\_{t} -\gamma\nabla_{\theta} \left[\mathcal{L}(g\_{\theta_t}(x_l),y_l)+\lambda\mathcal{L}(g\_{\theta\_{t}}(x_u),\hat{y}\_u|\phi\_{t})\right]$" represents a virtual future student, $\theta_{t+1}$, which means the teacher is only updated with the virtual $(t+1)$-step student with Eq. (1).
> This is the most straightforward version of FST.
> In the improved FST in Eq. (4), we update the teacher with *both* the $t$-step student and the $(t+1)$-step student, to help the teacher gain more knowledge.
> Your understanding is correct and we have exactly the same implementation as you have suggested.
> We have clarified this in the revised version.
>
>
>
> **Q2: About the implementation of FST-D.**
>
> At each iteration $t$, we first make a "copy" of the current student, $\theta_t$, and then conduct virtual exploration to obtain future states, *i.e.*, update the "copy" for $K$ steps under the supervision of the teacher.
> During such a virtual exploration, the teacher co-evolves with the "copy", while the original student, $\theta_t$, remains untouched.
> Finally, the advanced teacher is used to provide pseudo-labels for the original student, $\theta_t$, and perform updating *only once*.
>
>
>
> **Q3: Why can maintaining an ahead model save training time?**
>
> For instance, if we maintain an ahead student model $\theta'$, which is trained parallelly with the original student, $\theta$, but $K$ steps faster than $\theta$.
> We can directly obtain the virtual future model states from the ahead model $\theta\_{t+K}'$ to guide the current student training $\theta_t$.
> Since the ahead model is always $K$ steps faster than the student, we can *skip the virtual exploration*, and instead store the model weights $\theta'\_{t+1},...,\theta'_{t+K-1}$. That is how we trade space for time. We leave this as a future study as mentioned in the paper.

---

### Meta-Review · Area_Chair_KWZa · 2022-08-26

**Recommendation:** Accept
**Confidence:** Certain

**Metareview:**

This paper introduces an approach for reducing confirmation bias during self-training for semantic segmentation, by “learning from the future”, i.e. updating the teacher at a given timestep in self-training with a virtually updated version of the student, without actually using the gradients to update the student yet. Overall, reviewers were enthusiastic about the paper, finding the proposed method to be simple but interesting and of broad utility, and the paper well-written. The rebuttal responses seemed to address most questions and concerns, though there are some remaining weaknesses, such as the fact that the approach adds additional time/computation cost while the performance advantage versus standard self-training decreases with additional training iterations. However, on the balance I agree with reviewers that the strengths of the paper outweigh the weaknesses and recommend acceptance.

**Award:**

No

---

### Decision · Program_Chairs · 2022-09-14

Accept